# Cancer-Associated Fibroblast: Role in Prostate Cancer Progression to Metastatic Disease and Therapeutic Resistance

**DOI:** 10.3390/cells12050802

**Published:** 2023-03-04

**Authors:** Martina Bedeschi, Noemi Marino, Elena Cavassi, Filippo Piccinini, Anna Tesei

**Affiliations:** 1BioScience Laboratory, IRCCS Istituto Romagnolo per lo Studio dei Tumori (IRST) “Dino Amadori”, 47014 Meldola, Italy; 2Department of Medical and Surgical Sciences (DIMEC), University of Bologna, 40126 Bologna, Italy

**Keywords:** cancer-associated fibroblasts, prostatic neoplasm, tumor microenvironment, fibroblasts, reactive stroma, targeted therapy

## Abstract

Prostate cancer (PCa) is one of the most common cancers in European males. Although therapeutic approaches have changed in recent years, and several new drugs have been approved by the Food and Drug Administration (FDA), androgen deprivation therapy (ADT) remains the standard of care. Currently, PCa represents a clinical and economic burden due to the development of resistance to ADT, paving the way to cancer progression, metastasis, and to long-term side effects induced by ADT and radio-chemotherapeutic regimens. In light of this, a growing number of studies are focusing on the tumor microenvironment (TME) because of its role in supporting tumor growth. Cancer-associated fibroblasts (CAFs) have a central function in the TME because they communicate with prostate cancer cells, altering their metabolism and sensitivity to drugs; hence, targeted therapy against the TME, and, in particular, CAFs, could represent an alternative therapeutic approach to defeat therapy resistance in PCa. In this review, we focus on different CAF origins, subsets, and functions to highlight their potential in future therapeutic strategies for prostate cancer.

## 1. Introduction

Prostate cancer (PCa) is the second most prevalent malignancy in men worldwide, accounting for 7.8% of all cancers. Despite death rates declining since the 1990s due to early detection and advances in treatment [1], it is estimated that in 2020 there were around 1.41 million new cases globally [2]. Incidences vary considerably worldwide, but Northern Europe accounts for the highest number of cases with 83 per 100,000 males [2]. PCa is strongly associated with age, and the incidence increases sharply after the age of 50 [3]. Many patients present a slow development without any threat to mortality; yet, in several cases, PCa displays metastasis, develops aggressive subtypes, and carries a high mortality risk [4,5]. In 2020, PCa was the fifth most common cause of cancer death [2].

Androgen deprivation therapy (ADT), aimed at suppressing the aberrant activation of androgen receptor (AR) signaling, remains the standard of care for advanced prostate cancer [6,7,8]; instead, the treatment for metastatic castration-resistant prostate cancer (mCRPC) patients is based on chemo-radiotherapy regimens. Over the years, several drugs aimed at inhibiting circulating testosterone synthesis (abiraterone acetate), inactivating the AR (bicalutamide, enzalutamide), or hampering the growth and spread of prostate-metastatic cells (Cabazitaxel, Radium-223) have been approved by the FDA. Furthermore, several predictive biomarkers of response to anti-androgen or chemotherapy treatment have been suggested. Among these, a truncated splice variant of AR (AR-V7), lacking a ligand-binding domain, has emerged as an interesting predictive biomarker of resistance to abiraterone acetate and enzalutamide [9,10]. As emerged from the Lynch et al. study, family history is a significant risk factor for the development of PCa [11,12,13] and, in particular, BRCA2 mutations are responsible for an increased likelihood of early-onset of disease development and aggressiveness [14,15,16]. In addition, PTEN mutation or deletion in PCa has been demonstrated to be responsible for a worse disease outcome and a diminished response to abiraterone acetate [17,18,19]. Lastly, somatic gene fusions are recognized in several PCa patients, in particular, the TMPRSS2/ERG fusion seems to be responsible for taxane resistance [20,21].

The complexity of prostate cancer disease is shown by the fact that despite the range of drugs and therapeutic approaches developed until now, resistance to ADT or radio-chemotherapy arises from mechanisms still not fully understood; thus, there is an urgent need to develop new strategies to address this unmet medical need. Treatments currently in use mainly aim at tumor debulking or targeting AR; the latter representing a crucial oncogenic driver of prostate cancer. Approaches targeting immune components of the tumor microenvironment, such as immune checkpoint inhibitors (ICIs) (Sipuleucel-T) [4,6], have only recently been introduced into clinical practice to treat mCRPC patients (Figure 1a,b); however, the inability of immunotherapies to induce durable clinical responses to prostate cancer has highlighted the complex immunosuppressive nature of prostate cancer and its associated tumor microenvironment (TME) [22]. Despite the low number of research articles on the development of therapies targeting the main components of TME compared to that targeting AR signaling (Figure 1a,b), recent studies emphasize the network of cellular and molecular components, which collectively enable prostate cancer to maintain a complex immunosuppressive nature and to develop resistance to immunotherapies [22]. Indeed, stromal populations of prostate TME can represent a novel actionable target in the fight against cancer. Among TME cellular components, i.e., fibroblasts, smooth muscle cells (SMCs), endothelium, immune cells, and nerves, a pivotal role could be ascribed to cancer-associated fibroblasts (CAFs), the most abundant cell type within the prostate TME [23]. In particular, strong evidence suggests that CAFs are involved in the tumorigenesis and progression of prostate cancer, primarily through contact-dependent and paracrine-signaling mechanisms shared between activated and normal fibroblasts surrounding prostate cancer cells [23]. Furthermore, CAFs act as a barrier preventing drug delivery to cancer cells, having a tumor-promoting effect in several solid tumors, including prostate cancer [24].

The present work focuses on the current knowledge of the characteristics and functions of CAFs in prostate tumorigenesis and their prognostic and therapeutic potential in prostate cancer.

## 2. Prostate Fibroblasts

The prostate tissue architecture comprises ducts with epithelial luminal and basal layers and the surrounding stroma tissue. Interaction between the epithelium and stroma in normal prostatic tissue maintains physiological homeostasis. In short, the normal human prostate epithelium is composed of secretory epithelial cells, basal cells, and neuroendocrine cells, which are joined to a basal lamina. The other side of the lamina contains the stromal population composed of fibroblasts, smooth muscle cells, matrix, endothelial, and immune cells [25]. Fibroblasts and epithelial cells interact with one another, and this cooperation is essential during embryogenesis and organ development and for preserving epithelial and stromal differentiation [26]. For example, it is known that smooth muscle cells stimulated by the androgen in the prostate, induce the correct differentiation of epithelial cells by releasing regulatory molecular factors [26].

Fibroblasts composing normal prostate stroma are constituted by “resting” fibroblasts and two different populations of fibroblasts (Sca1+/CD90− and Sca1+/CD90+) able to respond to various physiological and pathological stimuli by adapting their phenotype and behavior. In particular, following tissue damage, resting fibroblasts undergo a phenotypic differentiation towards contractile myofibroblasts and start releasing several cytokines and chemokines in the local microenvironment, including tumor growth factor-β (TGFβ) and vascular endothelial growth factor A (VEGFA) [27,28,29,30,31]. Interestingly, fibroblast growth is also influenced by oxygen tension. Some studies found that hypoxia causes a delay in the wound-healing process due to reduced collagen production [32,33]. There are no universal markers for fibroblast, and they are generally characterized by the lack of epithelial, endothelial, and hematopoietic markers besides their morphology and localization within the tissue.

Following several noxious stimuli, similar changes also seem to take part in the development of prostate cancer [26,27,34].

## 3. Prostate Cancer-Associated Fibroblasts (CAFs)

In the face of tissue injuries or microbial infections, the homeostatic regulation capacity of the stroma can be reduced by processes associated with aging. In these circumstances, the stromal cells can secrete proinflammatory cytokines, such as CXCL12 and CXCL5 [35]. Furthermore, during aging, significant molecular and structural changes occur, besides the up-regulation of proinflammatory cytokines and growth factors, including the disruption of matrix components, increased trafficking of inflammatory cell types that may contribute to the pathological processes of benign prostate hyperplasia, prostatitis, and prostate carcinoma [36]. In the context of epithelial neoplasia, the prostate stroma undergoes phenotypic changes with a loss of well-differentiated smooth muscle cell population and the expansion of cancer-associated fibroblast (CAF) populations [36,37]. This set of alterations, termed “reactive stroma”, further co-evolves with tumor progression and is characterized by the presence of “myofibroblasts-like” CAFs, altered ECM deposition, neovascularization, and immune cell infiltration, similar to a wound healing niche (Figure 2) [38].

Furthermore, it was shown that CAFs communicate with prostate cancer cells, altering their metabolism and sensitivity to drugs [37,39]. Ippolito and colleagues demonstrated that CAFs could transmit molecular and metabolic inputs to cancer cells leading to mitochondria reshaping [40]. The mechanisms and pathways involved in this process are still under investigation, and several in vitro and in vivo studies have been carried out.

Similarly, the mechanisms underlying the interactions between PCa cells and CAFs are still not fully understood. It was hypothesized that CAFs could support metabolic cancer cell development and viability and foster cell stemness and metastasis spreading [41]. In support of this, the literature reported that CAFs derived from PC tissue, unlike normal fibroblast, enhanced the growth and tumorigenicity of either human benign prostatic cell line (BPH-1) or prostate cancer cell lines (LNCaP). These data, together with the observation that PCa stromal cells do not generally harbor somatic genomic mutations, strongly support the hypothesis that normal prostatic fibroblasts and CAFs interact differently and establish relationships with epithelial tumor cells, which, in turn, influence stromal cells, generating a tumor-supporting environment; however, CAFs’ role has not yet been clearly defined. This is, perhaps, due to prostate CAFs heterogeneity.

The variety of gene expression and the overlap of markers with various other cell types suggest that different mechanisms contribute to the presence of activated fibroblasts in the tumor microenvironment. Consequently, the origin of CAFs has been debated. Various markers have been suggested to classify the CAF population, such as FAP, FSP-1, vimentin, and α-SMA, nevertheless, none of these are CAF-specific [42,43]. Furthermore, CAFs and other cells composing TME express and secrete molecules able to mediate immunosuppression, such as tumor growth factor β (TGF-β), programmed cell death protein and ligand 1 (PD-1 and PDL-1), vascular endothelial growth factor (VEGF), IL-6, IL-8, IL-10, and IL-23 [42,43]. However, the heterogeneity of gene expression in CAF populations and the lack of markers exclusive to CAFs pose challenges to the identification and isolation of CAFs [42,43]. The main sources of CAFs are described below.

## 4. Potential Sources of CAFs and Their Heterogeneity in Prostate Cancer

Several terms have been used to describe the most frequent class of cells in the tumor stroma, such as “myofibroblast”, “cancer-associated myofibroblast”, “reactive stroma”, etc. [44]. Generally, it has been widely recognized that CAFs could arise from multiple cell precursors, which vary between tissues as evidenced by the heterogeneity of surface markers expressed [1,43]. Currently, the origins of the different CAF subtypes are under debate [26], and various cell precursors have been suggested (Figure 3). Among these, it was hypothesized that CAFs could be derived from resident fibroblasts, which undergo a mesenchymal–mesenchymal transition [26] toward the activated state. This process is also possible thanks to the high presence of growth factors, TGF-β, and reactive oxygen species (ROS) in the TME [45]. Furthermore, it has recently been reported how Yes-associated protein 1 (YAP1) can convert normal fibroblasts into CAFs in the PCa TME and can support the functional cross-talk between tumor cells and CAFs. It was shown that YAP1 silencing in tumor stromal cells could inhibit PCa growth [46].

Another source of CAFs may result from bone-marrow-derived mesenchymal stem cells (MSCs), which are recruited from distant organs using TGF-β or CXCL16/CXCR6-mediated chemotaxis [47,48]. Once in the tumor stroma, they express α-SMA, FAP, and other CAF-like phenotype markers [1,26,49]. In support of this, it has been demonstrated that MSCs can promote PCa cell growth and invasion in vitro and in vivo [1,26,49].

Several studies are currently demonstrating the role of senescent stromal cells in the creation of CAFs. Even though senescence is a helpful process used by cells to avoid dangerous mutations and uncontrolled proliferation, in the cancerous stroma, it is employed to boost tumor progression, for example, by expressing proinflammatory chemokines [26,50]. Indeed, they can activate epithelial cells, develop angiogenesis, and express α-SMA [50,51,52]. In PCa, senescent stroma plays a key role by promoting prostate cancer aggressiveness and mediating several processes ascribed to CAFs, such as vessel formation and inflammatory cell recruitment [51].

It was hypothesized that prostate CAFs could also come from endothelial cells. Indeed, several studies showed that mesenchymal–endothelial transition happens under the influence of TGF-β, inducing a change in cell morphology [53,54]. Over the course of the process, endothelial cells express FSP-1 marker, invasive characteristics, and lose cell–cell junctions [26,54]. Furthermore, endothelial cells have provided evidence of promoting PCa cell autophagy and cancer invasion. The mechanism was via suppression of both AR expression and AR transcriptional activity. It is possible that CCL5, secreted by endothelial cells, promotes tumor metastasis, even if further studies are required to verify the findings [55].

Lastly, adipose tissue and pericytes may also be potential sources of CAFs. The former can offer a propitious niche for tumor growth, probably via the paracrine mechanism and by inducing the production of ROS [56]. It has been reported that periprostatic adipose tissue produces paracrine factors that may regulate metalloproteinase activity and, in this sense, promote cancer aggressiveness and proliferation [57]. About pericytes, they show a phenotype similar to CAFs having in common several surface markers; however, further research is necessary to understand if they could be considered precursors of a distinct subpopulation of CAFs [26].

Regardless of origin, the transition to CAF is largely irreversible, and yet it remains plastic with regard to the CAF phenotype within or across tumor types. Since their involvement in promoting cancer growth and progression, all the prostate CAF subpopulations should be considered attractive therapeutic targets to reduce prostate cancer progression. For these reasons, it is important to highlight their potentialities and understand better their role, how they are recruited, and which factors are involved.

## 5. Role of CAFs in the Prostate Tumor Microenvironment and Cancer Progression

A further level of heterogeneity causing the identification of CAFs to be even more complex is caused by the different functions that CAFs perform in the TME (Figure 4). CAFs can be divided into three main subgroups, depending on their role in the TME: myofibroblastic CAFs, antigen-presenting CAFs (apCAFs), and immune-regulatory CAFs [58].

**Prostate CAFs and the Extracellular Matrix**. The extracellular matrix (ECM) is comprised of several molecules forming a three-dimensional reticulum able to support and influence stromal functions. Its remodeling is finely regulated; therefore, the loss of an organized matrix can be considered a distinctive sign of tumors and metastasis. In this landscape, CAFs are central players [58,59,60]. These CAFs called myofibroblastic CAFs due to their acquired contractile characteristic and high expression of α-SMA. They are involved in several processes of ECM remodeling, such as collagen deposition, smooth muscle contraction, and cancer cell growth and metastasis [58]. Several studies have demonstrated their presence in various organs, including the pancreas, breast, colon, and lung, showing their ability to localize close to the neoplastic cells. They also play a role in the early stages of tumor processes [58,61,62], particularly in the prostate, where they overproduce hyaluronate, fibronectin, and collagens, inducing morphological alteration and increased stiffness in the ECM. A recent study demonstrated that prostate CAF proteome is enriched for the “extracellular matrix” functional category, encouraging the assumption that CAFs modify the arrangement of reactive stroma and control prostate tumor cell motility, generating a tumor-supporting environment [23,63].

**Prostate Antigen-Presenting CAFs (apCAFs)**. Previously, CAFs were shown to suppress immune responses in the tumor microenvironment [64]. Accordingly, the recently identified MHCII+ antigen-presenting cancer-associated fibroblasts (apCAFs) in pancreatic adenocarcinoma (PDAC) and breast carcinoma (BC) are presumed to induce immune tolerance [61,62,65]. In contrast with the above-reported papers, the recent work by Kerdidani et al. showed that in situ MHCII antigen presentation within the tumor environment by a specialized subset of CAFs contributes to tumor immunity. The authors, pointing to mesenchymal cells as key partners in cancer therapy, envisage the in vivo delivery of tumor antigens to apCAFs (in vivo vaccination) as a possible scenario for the immunotherapeutic exploitation of their findings [66]. Further studies are needed to explore the presence of apCAFs also in PCa.

**Prostate Immune-Regulatory CAFs**. The lack of immunotherapeutic efficacy in prostate cancer, particularly in mCRPC, is partly due to the immunosuppressive TME that promotes the survival of the cancer cells and the progression of the disease [67]. Several studies reported that myeloid-derived suppressor cells (MDSCs) are a prominent immune cell subset infiltrating the castrate-resistant prostate cancer microenvironment [68,69,70,71]. In particular, macrophages, traditionally dichotomized into either classical (M1) or alternative (M2) phenotypes, are one of the most prominent immune cell populations in prostate cancer. A preclinical study described a complex web of relationships between CAFs, M2-polarized macrophages, and PCa, where the intersection between stromal and cancer cells sets up a pro-inflammatory microenvironment due to the enrichment in reactive CAFs and M2-like macrophages. Consequently, the stromal components play a significant role in PCa cell invasiveness, ultimately fostering cancer cells escaping from primary tumors and metastatic spread [72].

Another immune cell type well represented in the prostate TME is natural killer cells. It is known that resident fibroblasts are involved in preserving tissue homeostasis and in regulating the immune response. As in tumors, CAFs can produce chemokines and cytokines to recruit regulatory T cells and to change macrophage polarization; furthermore, they withdraw the cytotoxic activity of natural killers (NKs) by producing matrix-metalloproteinases (MMPs) capable of blocking NK effects [73].

Hitherto, several studies using scRNA-seq have described immune-regulatory CAFs heterogeneity, highlighting the differences between organs and pathologies. In particular, prostate CAFs mainly present CCL2, CXCL12, and IL33 markers as central features and play a pivotal role in regulating myeloid cell recruitment [58,74].

## 6. CAFs and Prostate Tumor Progression

Generally, CAFs are involved in tumor cell growth, invasion, epithelial-to-mesenchymal transition (EMT), and ADT resistance in PCa [75]. CAFs may promote tumor invasion through direct actions, such as cell-to-cell contact, stimulating cancer cell motility through the modulation of Eph–Ephrin signaling [76]. It has also been hypothesized that direct contact between PCa cells and CAFs stimulates tumor growth through Notch signaling in stromal cells [77]. CAFs can also exert indirect actions, such as the overproduction of ECM components (collagen, tenascin-C, fibronectin, or hyaluronic acid) [78,79,80], enhancing tumor cell proliferation and invasion, leading to metastasis [81,82]. In particular, it was reported that the fibronectin produced by CAFs could establish a fiber-oriented web fostering migration pathways of cancer cells [83]. A recent study performed using atomic force microscopy showed that prostate CAF contractile forces might alter the organization and the physical properties of the healthy prostate TME, making it permissive to tumor invasion [84]. In addition, it was reported that prostate CAFs induce ECM remodeling by secreting matrix metalloproteases (MMPs). In particular, during PCa development, the overexpression of MMP-1, -2, -7, -9, and -14 was found in stroma and circulation [85,86], as well as an imbalance between MMPs and tissue inhibitors of matrix metalloproteinases (TIMPs), which enhances PCa cell invasiveness [87,88]. It has been proposed that the loss of Dickkopf-3 (DKK3) expression, a secreted protein that inhibits TGF-signaling activity, either in prostate epithelial or stromal cells, could induce an increment of the expression of MMPs in PCa and the release of MMP-2 and MMP-9 [89,90].

Lastly, the literature reported that androgens could stimulate DNA synthesis in CAFs and shift their phenotype to a migratory one [91,92]. In particular, in PCa preclinical models was demonstrated that at low androgen concentration, the androgen receptor binds Src and p85α, leading to augmented MAPK and Akt activity, D1 upregulation, and boosted cell proliferation [91,92]. By contrast, at higher androgen concentrations, the stromal androgen receptor joins with Filamin A (FlnA) and integrin-β 1, activating Rac and promoting cell motility; thus, androgens induce fibroblast DNA synthesis at picomolar concentration while causing cell migration at higher concentrations [91]. Furthermore, Donato and colleagues demonstrated that the ARthat AR/FlnA complex regulates the androgen-challenged mobility and invasiveness in CAFs expressing androgen receptors. They used a 3D model of prostate cancer to show that the CAFs increased PC organoids dimension, further enhanced by androgens. To confirm, they showed that a peptide perturbing the AR/FlnA complex could reduce the organoid size, diminish biochemical alterations in ECM architecture, and impair CAFs’ migration toward PC cells [93]. All these findings suggest that androgens significantly influence CAFs and that the AR/FlnA complex may be a promising target for PCa treatment.

## 7. Prostate CAFs and Therapeutic Resistance

Growing evidence has shown the role of fibroblasts in prostate cancer initiation, progression, metastasis dissemination, and immune evasion [3]. In recent years numerous studies have demonstrated that prostate TME components, especially CAFs, could play a causal role in PCa therapy resistance. In particular, CAFs interact functionally with prostate cancer cells through direct contact or paracrine activity. Consequently, tumor cells secrete growth factors that promote tumor development, and CAFs alter ECM deposition and produce VEGF, which boosts abnormal cancer vascularization. In addition, they induce a more stem-like phenotype in PC cells, further increasing immune evasion and resistance to therapies [94]. IL-6, produced either by tumor prostate cells or CAFs, is another fundamental player within the PCa microenvironment that influences many aspects of prostate tumorigenesis, including insensitivity to androgens [95,96]. These observations led to the investigation of clinical use in PCa of inhibitors of IL-6 or its related transcription factor STAT3 [95,96].

Furthermore, several data showed that CAFs are sensitive to changes in tumor androgen levels, mediating the development of resistance to anti-androgen therapies. Indeed, AR signaling is active in CAFs, similar to smooth muscle cells and fibroblasts in the normal prostate stroma [97]. In particular, it was shown through Chip-seq studies that AR in CAFs interacts with different genomic sites than in PCa cells after testosterone exposure. AR in CAFs suppresses the expression of inflammatory cytokines, such as CCL2 and CXCL8, which have tumor-promoting properties [98]. Consequently, reduced AR signaling activity following ADT increases CAF-mediated secretion of these cytokines, enhancing PCa cell motility [98].

In addition, in 3D co-culture models of CAFs/PCa cell lines, it was demonstrated that CAFs decrease the sensitivity to the anti-androgens bicalutamide and enzalutamide through PI3K/Akt pathway [37]. Finally, it was reported that CAFs also have a negative impact on the effectiveness of general chemotherapies. For example, CAFs mediate resistance to cytotoxic agents, such as docetaxel, tuning the expression of a spectrum of genes, including the Wnt family member WNT16B, which promotes EMT in PCa cells [99]. CAFs also cause resistance to genotoxic agents, such as doxorubicin, by releasing glutathione, which reduces ROS levels and prevents drug accumulation in cancer cells, altering the mechanisms of cellular drug efflux and/or influx [100].

In recent years, the scientific community has shown greater interest in the fibroblast growth factor (FGF)/FGF receptor (FGFR) signaling pathway as it is involved in multiple tumor growth, development, and chemo-resistance [101]. Type 1–4 of FGFR is expressed in several cancer cells, including prostate cancer [102]. In addition, Hegab et al. demonstrated that FGF2 is overexpressed in fibroblasts close to the tumor area in lung cancer and suggested its role in supporting the tumor [101]. A recent study highlighted the ability of the insulin-like growth factor binding protein (IGFBP) 7, highly expressed in CAFs, to positively regulate FGF2 expression and release in gastric cancer. Consequently, it promotes tumor progression through the axis FGF2/FGFR1 [103].

Various studies have demonstrated their role also in PCa promotion, progression, and increased invasiveness [104,105]. In fact, in PCa, cells expressing high levels of FGFR1 increase metastases and overcome the inhibition of cell growth dependent on AR. A recent report displays that the activation of the FGFR1 pathway happens later in the disease’s development and mediates resistance to treatment [106]. In addition, another interesting study indicates that the expression of LIM domain only 2 (LMO2), which usually is suppressed by AR in healthy prostate fibroblasts, is overexpressed in PCa fibroblasts (CAFs) after ADT. Such a finding is relevant as the overexpression of LMO2 induces the paracrine release of FGF9 in TME able to promote tumor growth [107]. These prometastatic effects of FGFR1 open new prospects for a precision-targeted therapy against the FGF/FGFR axis in PCa. A recent phase II study combined an FGFR inhibitor (erdafitinib) with ADT. Although the results were promising thanks to the possible enhanced anti-tumor effect, the low tolerability in patients precludes clinical use [108].

Recently, research has focused on fibroblast activation protein (FAP), which is highly expressed in CAFs but not in normal fibroblasts. FAP is associated with a weak prognosis in several tumors and represents an interesting pharmaceutical target and biomarker.

FAP inhibitors (FAPIs) can be used as radioactive compound carriers delivered specifically to the tumor area. FAPI-04 was seen to be a successful theranostic tool giving results for the diagnosis and also promising outcomes for therapy [109,110].

## 8. Conclusions

Prostate cancer is the fifth most common cause of tumor-related death in men. The FDA has approved numerous drugs to treat the disease; however, ADT remains the standard therapy, even if patients invariably develop drug resistance and metastasis. In recent years, CAFs’ role in tumor development has become more evident, and, for many neoplasias, it is now a common opinion that targeting them could be an interesting therapeutic approach. In PCa, CAFs are the most abundant cell type, and strong evidence is emerging of their role in the tumor and how they could greatly influence cell growth, invasion, and ADT resistance. Among these, the findings indicate that CAFs expressing AR are sensitive to changes in androgen levels that tune their mobility and invasiveness and the properties of CAFs to alter the drug extrusion of epithelial cells through glutathione release; therefore, to overcome therapy resistance in PCa, combinatorial therapies which consider the response of stromal cells to anti-androgens and chemotherapeutics are needed. However, the heterogeneity of gene expression in CAF populations and the lack of markers exclusive to CAFs pose challenges for a therapy tailored to CAFs. In this field, further studies are needed to explore the potentialities of prostate CAFs as therapeutic targets.

## Figures and Tables

**Figure 1 cells-12-00802-f001:**
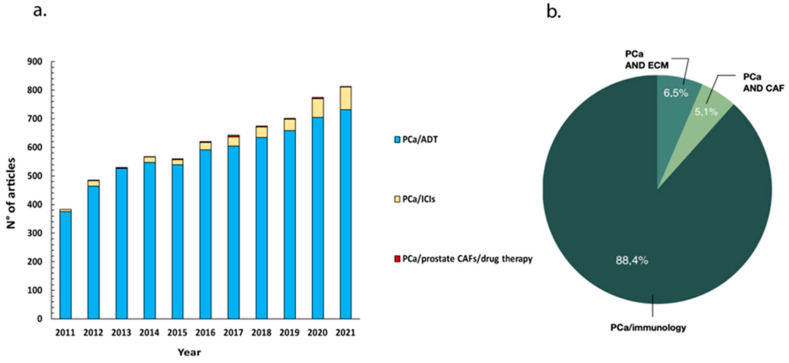
Therapeutic strategies targeting prostatic stromal microenvironment. (**a**) Search for articles appearing in PUBMED over the past 10 years (2011–2021) using the mesh terms “prostatic neoplasm/prostate CAFs/drug therapy” (red); “prostatic neoplasm/ICIs” (yellow); and “prostatic neoplasm/ADT” (blue). In the last 10 years, publications regarding therapies targeting the major components of TME have developed. (**b**) Search for articles appearing in PUBMED using the mesh terms “prostatic neoplasm” AND “extracellular matrix” (6%), “prostatic neoplasm” AND “cancer-associated fibroblast” (5%), and “prostatic neoplasm/immunology” (88%). Among PCa therapies against the major components of TME, immune therapies represent the most frequently searched, indicating a lack of research on CAFs in PCa.

**Figure 2 cells-12-00802-f002:**
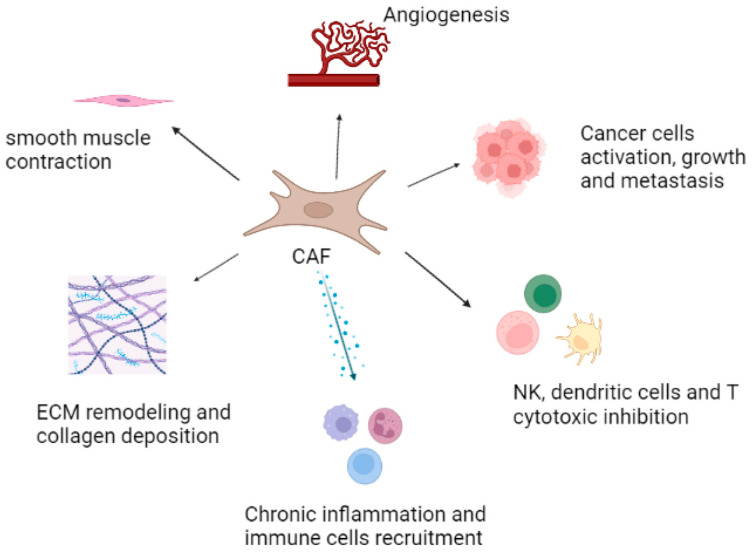
Features and functions of CAFs in TME. The major roles played by CAFs are represented in the figure: CAFs promote angiogenesis, induce smooth muscle contraction, stimulate ECM remodeling and collagen deposition, and boost cancer growth and metastasis, chronic inflammation, and immune inhibition.

**Figure 3 cells-12-00802-f003:**
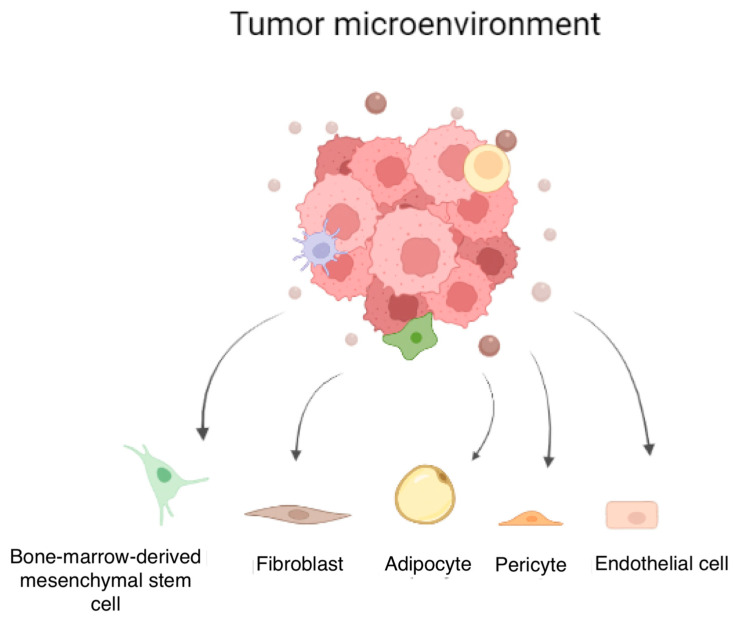
Potential sources of CAFs. In PCa, TME interacts with proximal and distal cells, such as bone-marrow-derived mesenchymal stem cells, healthy fibroblasts, adipocytes, pericytes, and endothelial cells, inducing the switch into CAFs. CAFs build paracrine communication with cancer cells by releasing factors, such as IL-6, IL-8, TGFβ, FGFs, VEGF, and GDF15, stimulating tumor growth, angiogenesis, and progression.

**Figure 4 cells-12-00802-f004:**
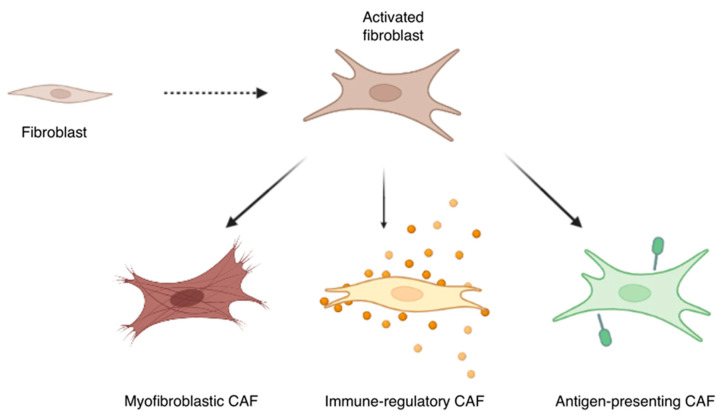
TME produces several stimuli able to induce fibroblast activation. CAFs can be divided into three substantial subpopulations: myofibroblastic, immune-regulatory, and antigen-presenting (apCAFs) CAFs. The first ones are responsible for the reorganization of the ECM, producing hyaluronate, fibronectin, and collagens and inducing morphological alteration and increased stiffness. Immune-regulatory CAFs and apCAFs are involved in cancer inflammation and the modulation of immune responses in the tumor microenvironment.

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
