# Peer review of "Cancer-Associated Fibroblast: Role in Prostate Cancer Progression to Metastatic Disease and Therapeutic Resistance"

_cells, 2023, doi:10.3390/cells12050802_

Round 1

Reviewer 1 Report

This is an interesting review regarding tumor microenvironment and cancer associated fibroblast associated with prostate cancer. I think it is well written and provide good inforation for authors. However, there are some confudes style of references in the number of 5, 10, 17, 76, and 85. I suggest carful check the reerence structure.

1. The main question addressed by the authrors is tumor microenviroment in which tumor associated fibroblast may influence the progress of prostate cancer and may as a therapeutic target.

2. The topic is a brand review regarindg prostate cancer with its microenviroment that may be an alternative treatment option beyond androgen deprivation therapy cause it may develop castration resistant.

3. What does it add to the subject area compared with other published
material?  -> I think it is OK.

4. What specific improvements should the authors consider regarding the
methodology? What further controls should be considered?  -> This is a review article. Therefore, no furhter controls needed.

5. Are the conclusions consistent with the evidence and arguments presented  and do they address the main question posed?  -> Yes

6. Are the references appropriate? -> I have mention the references should be corrected.

7. Please include any additional comments on the tables and figures. -> The figures are plotted well. 

Reviewer 2 Report

Dr. Anna Tesei and co-authors submitted a mini-review manuscript for review entitled “Cancer-Associated Fibroblast: Role in Prostate Cancer Progression to Metastatic Disease and Therapeutic Resistance”. This manuscript deals with the field of experimental-descriptive science. It provides a detailed overview of the basics of the formation, development and treatment of prostate cancer, therefore, this manuscript may be published in the international journal "Cells" after removing the minor correction errors listed below:

There are a lot of well-defined abbreviations in the manuscript, used many times throughout the text. Cells is aimed at a wide range of readers. To make it easier for them to study the manuscript, please add a "Abbreviations" section.

On line 1 is: ... Review ... but it would be better ... Mini-review ... . Comment: The content of this rather short and concisely written manuscript consists of 9 pages of print with pictures.

Please synchronize the style of titles/names of chapters 1–7 and conclusion. Authors use capitalization style in names (at Paragraph 2–5, and 7), other times authors use the normal naming style (Paragraph 1, 6, and Conclusion). At the end of the paragraph title number 6 is a full stop, and the other paragraph titles are missing a full stop.

At line 90 is: … PCa. . … , should be … PCa. … . Comment: One dot too many has been entered.

At line 184 is: … ,46] … , but should be … ,46]. … . Comment: Please add dot mark.

At line 363 is: … 5. Conclusions … , but should be … 8. Conclusions … . Comment: Please correct the numbering of the paragraphs.

On lines 380–382, please complete the information or delete the patent section/paragraph.

At line 420 at the authors name list please delete the sentence “ The UKGPCS Collaborators; “.

In line 434, is the content of cross-reference number 19 correct? Provide more detailed article, book or encyclopedia information with ISBN or doi number, year of publication, volume, pages as needed. Comment: Please check the correctness of the automatically entered references, point by point.

Reviewer 3 Report

The article addresses a very complicate item- such as tumor micro-environment TME. This is a very interesting approach - fibroblast, responsible for healing process but also for failure in oncology- decreasing tissue oxygen and diminished arrival of different drugs at cellular level. Maybe these aspects could be developed somewhere in the article, also some aspects regarding FGFR and targeted therapies.

Reviewer 4 Report

Tesei et al reviewed role of Cancer-Associated Fibroblast in Prostate Cancer Progression to Metastatic Disease and Therapeutic Resistance. This is an interesting article. It has worth to be published but need to revision in English because there are several poor structured sentences.
